# Gynecologic health of women with multiple sclerosis: An overview on the current status and findings of Pap tests in a low-income setting

**Masoud Etemadifar[1,2], Shima Shoeib[2,3,4], Mehri Salari[5], Mohammadreza Etemadifar[3], Nahad Sedaghat[2,3,4,6]***

1 Department of Neurosurgery, School of Medicine, Isfahan University of Medical Sciences, Isfahan, Iran, 2 Alzahra Research Institute, Alzahra University Hospital, Isfahan University of Medical Sciences, Isfahan, Iran, 3 School of Medicine, Isfahan University of Medical Sciences, Isfahan, Iran, 4 Student Research Committee, Isfahan University of Medical Sciences, Isfahan, Iran, 5 Faculty of Medicine, Shahid Beheshti University of Medical Sciences, Tehran, Iran, 6 Student Research Committee, Kashan University of Medical Sciences, Kashan, Iran

* nahad.sedaghat@gmail.com

**Data availability statement:** All data generated within the study is reported in the present manuscript and/or associated files in aggregate form. Raw data cannot be shared publicly because of potentially identifiable information. Data are available from the Isfahan University of Medical Sciences Institutional Data Access

## Abstract

### Background

Women with MS (wwMS), particularly ones in low-income settings, and exposed to disease-modifying therapy (DMT), could have specific gynecological health-related issues.

### Aim

To assist policy making and lead further research by describing the current status of gynecological health and Pap test results in wwMS.

### Methods

Cross-sectional study on wwMS living in Isfahan, Iran. Participants were surveyed and referred for a Pap test, results of which were compared with 1:2 age- and socioeconomic status-matched healthy controls (HC). Primary outcome was the degree of non-benign squamous/glandular cell abnormalities. Secondary outcomes were presence of evidence of infection, and the degree of benign inflammatory/reactive changes. Logistic regression models were utilized for analyses.

### Results

197 wwMS were included (mean age [SD], 41.2 [8.3]; median EDSS (IQR) 1.5 [0.5]). 74.1% reported having sexual activity more than once per week in the past year. For contraception, 21.6% and 16.8% used calendar-based methods and male condoms, respectively. 7% had contracted a gynecological infection in the past. Only 1% had received HPV vaccination. Compared to HC, benign reactive/inflammatory changes in Pap tests were less frequently seen in the wwMS (OR: 0.3; 95% CI: 0.2, 0.4; $p < 0.001$), while evidence

/ Ethics Committee (contact via rec@mui.
ac.ir) for researchers who meet the criteria for
access to confidential data. Requests in this
regard could also be sent to the corresponding
author (contact via nahad.sedaghat@gmail.
com).

**Funding:** The author(s) received no specific
funding for this work.

**Competing interests:** The authors have
declared that no competing interests exist.

of infection was seen more frequently (OR: 11.5, 95% CI: 3.3, 40; $p < 0.001$). Results were consistent across DMT groups except anti-CD20 therapies. Additionally, the frequency of non-benign changes in wwMS was two times of that in the HC, but the study lacked adequate power to confirm statistical significance (1.5% vs. 0.8%, OR: 2; 95% CI: 0.4, 10.1; $p = 0.39$).

## Conclusion

There is room for improvement of the gynecological health status of wwMS who live in low-income settings. Also, findings support an immune dysfunction in the cervices of DMT-exposed wwMS. Additionally, further research is merited to determine the risk of changes of malignant potential in cervices of wwMS.

## Introduction

Female biological sex is among the risk factors for multiple sclerosis (MS) – a demyelinating CNS disease of unknown etiology [1,2]. Based on current guidelines, disease-modifying therapy (DMT) is indicated for chronic management of MS [3]. Different DMT regimens act through a wide variety of mechanisms of action (MOA); e.g., some prevent the trafficking of immune cells (e.g., fingolimod and natalizumab), some prevent the division and expansion of immune cells (e.g., teriflunomide), some divert the immune response from the CNS (e.g., glatiramer acetate [GA]), and some eliminate lymphocytes (e.g., anti-CD20 monoclonal antibodies [aCD20], cladribine, alemtuzumab) [4–6]. Regardless of the MOA, all of these treatments act on the immune system, potentially predisposing the women with MS (wwMS) to diseases of the female reproductive system, e.g., gynecological infections and cervical cancer. Notably, altered immune system function could hinder the clearance of human papillomavirus (HPV) infection. Currently, it is unknown whether the prevalence of HPV and/or the risk of its progression towards malignancy is elevated in wwMS compared to the general population [7,8]; but it is known that HPV is contracted by most sexually-active people of both sexes [9–11], and is involved in 90% of anal, 70% of vaginal, 50% of penile, 40% vulvar and 13-72% of oropharyngeal cancers [12–14].

The potential gynecological health-related issues specific to the wwMS, particularly in context of DMT exposure, could be properly managed if identified and addressed early. For instance, HPV vaccination could be offered to the wwMS if they are shown to bear an additional risk for HPV infection [15]. Other instances could be offering behavioral consultation, sanitary products, as well as routine screening for, and treating of infections in both wwMS and their partners, if they are shown to bear an additional risk for such diseases [16–18].

Since the worldwide adoption of the current MS treatment paradigms, there have been studies that have aimed to identify these issues in wwMS. Yet, the current literature lacks studies inclusive of the entirety of the wwMS, particularly, ones living in low-income settings. This is despite the fact that a considerable worldwide burden of gynecological health issues is imposed on people living in low-income settings [19].

Therefore, we intended to contribute to the evidence and lay a foundation for further research in this regard, by providing an overview on the current status of gynecological health among a group of DMT-exposed wwMS, as well as their Pap test results. Hereby, our study is reported, in accordance with the STROBE statement [20].

## Methods

### Population and setting

An observational study with a cross-sectional design was conducted on the wwMS seen at the Isfahan MS clinic, Isfahan, Iran, from January 2023 until April 2024. Participants were identified prospectively by their MS neurologist from ones referring to the clinic for routine follow-up visits. Following confirmation of eligibility (based on the criteria below), demographics and clinical information of participants were retrieved from the electronic medical records (EMRs) of the clinic. Later, participants were surveyed by a medical intern, and then referred to a single, prespecified laboratory for a liquid-based Pap test. At the laboratory, specified, trained nurse practitioners obtained Pap samples. The results of the Pap tests, as reported by the laboratory pathologist, were received and used for the study. Furthermore, Pap test results of wwMS were compared with a 1:2 age- and socioeconomic status (SES)-matched group of healthy controls (HC) who underwent Pap tests for cervical cancer screening in the same laboratory and in the same period.

In addition to being able and willing to provide a written informed consent, specific inclusion criteria for the wwMS comprised:

1. definitive diagnosis of MS, based on the 2017 revisions of the McDonald criteria [21], at least a year prior to recruitment;

2. having experienced receptive vaginal intercourse at least, three years prior to recruitment;

3. absence of any evidence (comprising any sign/symptom and/or paraclinical evidence) of any gynecological pathology at the time of recruitment;

4. having no history of any cancer and/or pre-malignant disease;

5. being equal or above the age of 21 years; and

6. absence of any medical (e.g., active menstruation/spotting within 48 hours, sexual intercourse within 48 hours, certain anatomical abnormalities, use of vaginal products within 48 hours, reported allergy/hypersensitivity to lubricants/specula, etc.), cultural (e.g., religious beliefs, mistrust in the performing provider, language barriers, etc.), and/or ethical (e.g., withdrawal of consent, confidentiality issues, etc.) contraindication for Pap testing.

Exclusion criteria for the wwMS comprised:

1. withdrawal of consent at any time; and

2. inadequacy of the obtained Pap sample for analysis, e.g., complete absence of endocervical cells in the visualized Pap smear.

Furthermore, apart from the MS diagnosis, the same eligibility criteria were considered for the HC.

### Variables and measurements

Information pertaining to the following variables were extracted from the EMRs of the clinic: age and duration of MS (measured in years), clinical course of MS (relapsing or progressive), current EDSS score, DMT type and duration of DMT. A medical intern obtained information regarding the following variables via interviews with the wwMS: SES (low, middle, or high-income, based on self-reported household income, classified according to the figures reported by the statistics center of Iran, available at https://amar.org.ir/cost-and-income), marital status, sexual debut date, average coital frequency in the past year (with the following

cut-offs: once per day, twice per week, once per week, once per month), current fertility/ menstrual status (premenopausal and fertile, premenopausal and not fertile, peri/postmeno-pausal), method used for contraception (e.g., calendar-based, barrier, hormonal methods, etc.), duration elapsed since first vaginal delivery date (if applicable), comorbidities, history of gynecological disease/infection, having familial history of any cancer, average number of menstrual pads used daily during menstruations, and usage of vaginal cleansing products. Information on the following variables were extracted from the reports of the Pap test results: presence and degree of inflammatory/reactive cellular changes (mild, moderate, severe), presence of infection (e.g., visualization of coccobacilli indicative of bacterial vaginosis, visu-alization of yeasts indicative of vaginal candidiasis, etc.), presence and degree of non-benign squamous and/or glandular cell abnormalities (atypical cells of undetermined significance [ASCUS], atypical glandular cells of undetermined significance [AGUS], low-grade squamous intraepithelial lesion [LSIL], high-grade squamous intraepithelial lesion [HSIL], atypical squa-mous cells, cannot rule out HSIL [ASCH], adenocarcinoma).

A descriptive overview of all variables was presented to demonstrate the current status of gynecological health among wwMS. For the comparative analyses, an ordinal scale compris-ing the presence and degree of non-benign squamous and/or glandular cell abnormalities constituted the primary outcome (grade 0: none; grade 1: ASCUS, AGUS; grade 2: LSIL, grade 3: HSIL, ASCH, adenocarcinoma.) Furthermore, a binary variable comprising presence or absence of evidence of infection, an ordinal scale comprising the presence and degree of inflammatory/reactive cellular changes (grade 0: none; grade 1: mild; grade 2: moderate; grade 3: severe) were considered as secondary outcomes. All other variables were considered as covariates.

## Statistical methods

Descriptive statistics were presented with appropriate measures such as mean and standard deviation (SD), count and percentage, etc. Comparisons and hypothesis testing were done with appropriate (non)parametric statistical tests. Univariate and multivariable, ordinal and binary logistic regression models were utilized to investigate the associations between covari-ates and the outcome variables. P value ≤ 0.05 was considered as the criterion of statistical significance.

## Ethics and data availability

Ethical approval for the present study has been obtained from the research ethics commit-tees (REC) of the School of Medicine, Isfahan University of Medical Sciences (Approval ID: IR.MUI.MED.REC.1401.217). Written informed consent was acquired from participants. All data generated within the study is reported in the present manuscript and/or associated files in aggregate form. Raw data cannot be shared publicly because of potentially identifiable information. Data are available from the Isfahan University of Medical Sciences Institutional Data Access/ Ethics Committee (contact via rec@mui.ac.ir) for researchers who meet the criteria for access to confidential data. Requests in this regard could also be sent to the corre-sponding author (contact via nahad.sedaghat@gmail.com).

## Results

### Characteristics of participants

Of ~ 250 wwMS considered for inclusion, 197 (78.8%) were surveyed and underwent the Pap test. Mean (SD) age of participants was 41.2 (8.3) years; median (IQR) duration since MS diagnosis was 9 (10) years. At the time of the study, 18 (9.1%) had secondary progressive,

while the others had relapsing-remitting MS. The median (IQR) EDSS score of participants was 1.5 (0.5). All participants were treated with DMT. More specifically, throughout their course of disease, 106 (53.8%) had underwent treatment with interferons (mean [SD] duration of exposure: 8.1 [5.2] years), 32 (16.2%) with GA (mean [SD] duration of exposure: 4.1 [2.8] years), 52 (26.4%) with dimethyl fumarate (DMF) (mean [SD] duration of exposure: 3.8 [2.8] years), 34 (17.3%) with teriflunomide (mean [SD] duration of exposure: 3.5 [2.1] years), 29 (14.7%) with fingolimod (mean [SD] duration of exposure: 5.0 [5.0] years), 3 (1.5%) with natalizumab (mean [SD] duration of exposure: 3.0 [2.6] years), and 33 (16.8%) with aCD20 (mean [SD] duration of exposure: 4.4 [3.7] years).

## Gynecological health status

The summary of the surveys on the gynecological health-related factors are presented in Table 1. As noted, most participants reported having sexual activity more than once per week in the past year, with calendar-based methods, male condoms, and partner vasectomy being the first, second, and third most common reported methods used for contraception, respectively. Almost all participants used at least, three menstrual pads daily during menstruation. Less than a fourth of participants reported using vaginal cleansing products. 14 (7.1%) of the participants reported contracting a gynecological infection in the past with an unknown pathogen. Additionally, only two participants (1%) reported undergoing HPV vaccination: one with two doses and one with one dose, both of a quadrivalent formulation.

## Pap test results

From Table 2 the summary of Pap test results could be interpreted. As seen, the observed benign reactive/inflammatory changes differed between the participants with MS and matched HC; such changes were less frequently observed in the wwMS (OR: 0.3; 95% CI: 0.2, 0.4; $p < 0.001$). Further of note, among all, only one of the wwMS with evidence of yeast infection showed severe benign inflammatory changes, whereas all HC with evidence of infection showed severe benign inflammatory changes. Interestingly, evidence of infection was more frequently observed in wwMS compared to the HC (OR: 11.5, 95% CI: 3.3, 40; $p < 0.001$). When stratified by DMT at the time of the Pap test, wwMS treated with interferons (OR: 10.6; 95% CI: 2.1, 54.2; $p = 0.005$), GA (OR: 12.4; 95% CI: 2, 78.3; $p = 0.007$), DMF (OR: 12.1; 95% CI: 2.6, 56.3; $p = 0.001$), teriflunomide (OR: 9; 95% CI: 1.4, 55.9; $p = 0.02$), fingolimod (OR: 24.4; 95% CI: 4.6, 130.7; $p < 0.001$), but not the ones treated with aCD20 (OR: 4.5; 95% CI: 0.5, 44.6; $p = 0.20$), showed evidence of infection more frequently than the HC (Fig 1B). The ones treated with interferons (OR: 0.2; 95% CI: 0.1, 0.4; $p < 0.001$), GA (OR: 0.3; 95% CI: 0.1, 0.7; $p = 0.004$), DMF (OR: 0.3; 95% CI: 0.2, 0.5; $p < 0.001$), teriflunomide (OR: 0.3; 95% CI: 0.1, 0.6; $p = 0.001$), fingolimod (OR: 0.4; 95% CI: 0.1, 0.9; $p = 0.04$), but not the ones treated with aCD20 (OR: 0.7; 95% CI: 0.3, 1.6; $p = 0.44$), also showed benign reactive/inflammatory changes less frequently (Fig 1A).

Additionally, the frequency of non-benign cellular changes in wwMS was two times of that in the matched HC (3/197 [1.5%] vs. 3/394 [0.8%], OR: 2; 95% CI: 0.4, 10.1; $p = 0.39$). Particularly, one case of adenocarcinoma (0.5%), one LSIL (0.5%), and one ASCUS (0.5%) were observed among the wwMS, while one case of LSIL (0.3%), and two ASCUS cases (0.5%) were observed among the HC. None of the wwMS with non-benign cellular changes were ever exposed to high-efficacy DMT (i.e., aCD20 or natalizumab), fingolimod or DMF. Particularly, the one with adenocarcinoma was exposed to teriflunomide for two years, the one with LSIL was exposed to interferons for 10 years and GA for 6 years, and the one with ASCUS was exposed to interferons for 7 years. The difference between the wwMS and HC in terms of the

**Table 1. Characteristics of participants, and results of the survey on the gynecological health and related factors among women with MS.**

| Variable | All participants with MS (n = 197) |
|---|---|
| Age (mean, SD) | 41.2 (8.3) |
| Marital status (n, %) | |
| - Married | 188 (95.4) |
| - Divorced | 5 (2.6) |
| - Other/ preferred not to disclose | 4 (2) |
| Duration since sexual debut (mean, SD) [years] | 19.3 (10.1) |
| Average coital frequency in the past year (n, %) | |
| - Less than once per month | 12 (6.1) |
| - Once per month to less than once per week | 39 (19.8) |
| - Once per week | 34 (17.2) |
| - Twice per week to less than once per day | 64 (32.5) |
| - Once per day or more | 7 (3.6) |
| - Preferred not to disclose | 41 (20.8) |
| Current fertility/menstrual status (n, %) | |
| - premenopausal and fertile | 180 (91.4) |
| - premenopausal and not fertile | 10 (5.1) |
| - peri/postmenopausal | 7 (3.5) |
| Method used for contraception (n, %) | |
| - None | 99 (50.2) |
| - Calendar-based | 40 (20.3) |
| - Male condoms | 31 (15.7) |
| - Female condoms | 1 (0.5) |
| - Oral contraceptive pills | 3 (1.6) |
| - Medroxyprogesterone acetate injection | 1 (0.5) |
| - Intrauterine device | 6 (3) |
| - Partner vasectomy | 13 (6.6) |
| - Other/ preferred not to disclose | 3 (1.6) |
| Duration since first vaginal delivery (mean, SD) [years] | 18.6 (9.8) |
| Comorbidities (n, %) | |
| - None | 134 (68) |
| - Diabetes mellitus | 7 (3.6) |
| - Hypertension | 10 (5.1) |
| - Hyperlipidemia | 12 (6.1) |
| - Smoking | 9 (4.6) |
| - Cardiovascular disease | 8 (4.1) |
| - Kidney disease | 8 (4.1) |
| - Liver disease | 4 (2) |
| - Lung disease | 2 (1) |
| - Thyroid disease | 22 (11.2) |
| - Cancer | 2 (1) |
| - Other (not specified) | 4 (2) |
| History of gynecological disease (n, %) | |
| - None | 170 (86.3) |
| - Ovarian cyst | 14 (7.1) |
| - Polycystic ovary syndrome | 7 (3.6) |
| - Other/ preferred not to disclose | 6 (3) |

*(Continued)*

**Table 1.** (Continued)

| Variable | All participants with MS (n = 197) |
|---|---|
| History of gynecological infection (n, %) | |
| - None | 183 (92.9) |
| - Yes, unknown pathogen | 14 (7.1) |
| Family history of any cancer (n, %) | |
| - None | 138 (70.1) |
| - Yes, in parent(s) | 17 (8.6) |
| - Yes, in sibling(s) | 6 (3) |
| - Yes, in grandparent(s) | 14 (7.1) |
| - Yes, in parents' sibling(s) | 16 (8.2) |
| - Yes, in cousin(s) | 6 (3) |
| Average number of menstrual pads used during menstruations (n, %) | |
| - 1 to 2 per day | 8 (4.1) |
| - 3 to 5 per day | 136 (69) |
| - 6 to 8 per day | 39 (19.8) |
| - More than 8 per day | 14 (7.1) |
| Usage of vaginal cleansing products (n, %) | |
| - No | 149 (75.6) |
| - Yes | 44 (22.4) |
| - Preferred not to disclose | 4 (2) |

Abbreviations: MS, multiple sclerosis; SD, standard deviation.

frequency of non-benign cellular changes was not classified as statistically significant, nevertheless, it should be noted that the present study was not designed to detect such difference with adequate statistical power.

Furthermore, using ordinal and binary logistic regression, the association of the pap test results (i.e., the outcome variables mentioned above) was investigated with the covariates age, SES, MS duration and subtype, total number of MS relapses, total number of times undergoing corticosteroid pulse therapy, EDSS score, historical and current DMT and duration thereof, and comorbidities. A positive association was found between current aCD20 therapy and the degree of benign inflammatory/reactive changes (unadjusted OR vs. interferons: 2.7; 95% CI: 1.1, 6.8; $p = 0.03$). This association was seen multivariable analysis as well (adjusted OR vs. interferons: 3.5; 95% CI: 1.3, 9.1; $p = 0.01$). The association between comorbidities and non-benign cellular changes could not be investigated as none of the three wwMS with non-benign cellular changes reported having a comorbidity. Other results returned unremarkable.

## Discussion

The present study provided an overview on the gynecological health status and related factors among wwMS living in a low-income setting. Findings could be used for future policy making in such settings, as well as laying a base for proper design and conduction of future studies. Although some findings may stem from an increased susceptibility of wwMS to gynecological infection and malignancy compared to the general population, larger studies are warranted to draw confident conclusions in this regard.

**Table 2. Summary of Pap test results among women with MS and age- and socioeconomic status-matched healthy controls.**

| Variable | Participants with MS (n = 197) | Healthy controls (n = 394) | P value |
|---|---|---|---|
| Benign inflammatory/reactive cellular changes (n, %) | | | <0.001 |
| - None | 58 (29.4) | 22 (5.6) | |
| - Mild | 76 (38.6) | 150 (38.1) | |
| - Moderate | 48 (24.4) | 188 (47.7) | |
| - Severe | 15 (7.6) | 34 (8.6) | |
| Evidence of infection (n, %) | | | 0.001 |
| - None | 181 (91.9) | 388 (98.4) | |
| - Yes, coccobacilli | 3 (1.5) | 2 (0.5) | |
| - Yes, other bacteria | 9 (4.6) | 3 (0.8) | |
| - Yes, yeasts | 4 (2) | 1 (0.3) | |
| Non-benign cellular abnormalities | | | 0.52 |
| - None | 194 (98.5) | 391 (99.2) | |
| - ASCUS or AGUS | 1 (0.5) | 2 (0.5) | |
| - LSIL | 1 (0.5) | 1 (0.3) | |
| - ASCH, HSIL, or adenocarcinoma | 1 (0.5) | 0 | |

Abbreviations: MS, multiple sclerosis; ASCUS, atypical cells of undetermined significance; AGUS, atypical glandular cells of undetermined significance; LSIL, low-grade squamous intraepithelial lesion; HSIL, high-grade squamous intraepithelial lesion; ASCH, atypical squamous cells, cannot rule out HSIL.

Earlier studies generally indicated that the risk of cervical cancer is not increased in wwMS compared to the general population [22,23]. Yet, adoption of the recent MS treatment paradigms may have changed the circumstances. Furthermore, although we could not find any evidence indicating a different presenting symptomatology of cervical cancer in wwMS compared to general population, a delay in diagnosis may be plausible as presenting symptoms may be attributed to MS-related issues; thus, more extensive gynecologic check-ups were recently recommended for wwMS [7]. Moreover, it is currently not known whether HPV infection and/or its progression towards invasive disease are more prevalent in wwMS compared to the general population [7,8]. Yet, recent studies are documenting an increase in the detection of cervical pathologies of malignant potential in wwMS, which could be attributable either to the use of newer DMTs or to more extensive check-ups per recent recommendations [7,8]. For instance, a recent registry-based Australian longitudinal cohort study [24] on 248 wwMS from 1998 to 2019, reports that exposure to DMT is associated with a 3.5-fold increased risk of cytological and/or histological cervical pathology of malignant potential. In a Norwegian study [25] an increase is observed in female genital organ cancers in the period of adoption of new DMTs. Interestingly, a recent Mendelian randomization study on the UK biobank cohort found an association between MS and related factors (e.g., use of DMT) with cervical cancer but none of the other 14 investigated cancers [26]. In line, we observed that the odds of cytological cervical changes of malignant potential were two-folds in the DMT-exposed wwMS compared to matched HC. Yet, it should be noted, the present study was not designed to detect such difference with adequate statistical power. Despite that, results could be used to guide the design and conduction of future studies with adequate statistical power.

Furthermore, the present study showed a > 10-folds increased risk of observing cytological evidence of infection in DMT-exposed wwMS compared to HC. But interestingly, the risk

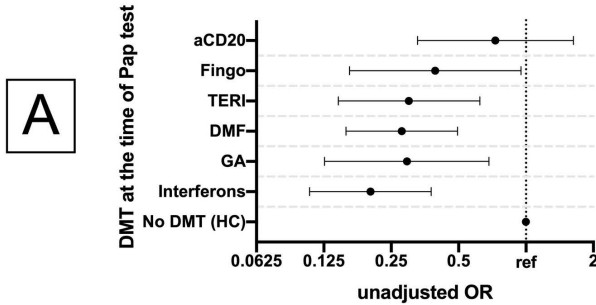

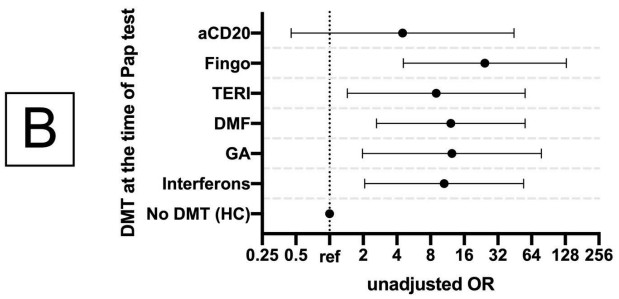

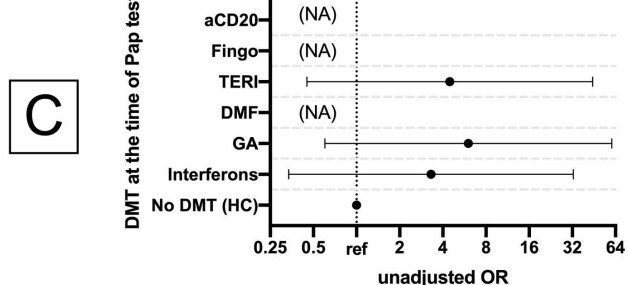

**Fig 1. Impact of DMT at the time of Pap test on its results.** The association between DMT at the time of the Pap test with the results of the Pap tests is illustrated to facilitate a comparison of the DMTs in terms of their effect on the outcomes. Error bars represent the 95% confidence intervals for the odds ratios. Abbreviations: aCD20, anti-CD20 therapy; DMT, Disease Modifying Therapy; Fingo, Fingolimod; GA, Glatiramer Acetate; HC, Healthy Controls; MS, Multiple Sclerosis; NA, not applicable; OR, Odds Ratio; ref, Reference; TERI, Teriflunomide; DMF, Dimethyl Fumarate.

of benign inflammatory/reactive cytological changes was reduced by 70% in DMT-exposed wwMS compared to HC. These results may be explained by an impairment in the mucosal immune response to cervical pathogens, possibly mediated by DMT; this still remains to be investigated as we could not find any study directly linking DMT to impairments in cervical mucosal immune responses to pathogens. Observation of lack of, or lower grades of such benign inflammatory/reactive changes in wwMS with evidence of infection, compared to severe grades of such changes in all HC with evidence of infection strengthens this hypothesis.

Furthermore, in the present study, higher-grade benign inflammatory changes were more frequent in wwMS treated with aCD20 – a DMT class acting mainly on B cells. This may suggest that an impairment in the T cell-mediated and innate immune functions underlies the dampened immune response to cervical pathogens. Further characterization of the processes underlying the lower grades of benign inflammatory changes despite higher rates of infection in DMT-exposed wwMS compared to HC is an interesting subject for future studies. Another explanation could be the impairment of afferent neural pathways in wwMS. Particularly, impairment of the neural pathways involved in sensory input from the cervix/vagina in the wwMS [27,28] could render their infections asymptomatic, and therefore, making them more likely to be included in the study than the HC with infection, whose intact afferent neural pathways render their infections symptomatic. This as well, could be a subject of merit for further studying.

Although the present study lacked the statistical power to determine whether wwMS are at increased risk for cervical cancer, it documents increased infections rates and decreased immune reactions to pathogens in the cervices of wwMS compared to HC. Proper immune system functioning is an important contributor to HPV clearance as documented in prior literature [29]. Furthermore, infections, e.g., with *Chlamydia trachomatis* may increase the risk of HPV progression towards malignancy [30]. Thus, the present study could support adoption of more stringent measures in prevention of HPV infection in this population, especially as HPV vaccination has been done in only 1% of the wwMS as we observed.

Lastly, the results of the present study should be interpreted in the context of important limitations. As mentioned, the study was not adequately sized to enable determination the cervical cancer risk in pwMS compared to the general population. Moreover, the people with progressive subtypes of MS were not represented well in the study, thus, limiting the generalizability of the results to this subset of pwMS, and increasing the possibility of missing a potential correlation between the MS subtype and the outcomes. Furthermore, information pertaining to important potential confounding factors (e.g., the number of prior pregnancies) was not collected, and the information pertaining to other survey variables were based on subjective reports of participants themselves (e.g., presence/absence of comorbidities such as diabetes were not objectively confirmed.) Moreover, obtaining of Pap samples has been done by trained nurse practitioners; obtaining of the samples by gynecologists could ensure the robustness of the results in future studies. Additionally, unlike the pwMS who were directly surveyed, the data pertaining to the HC group of the present study has been collected in a retrospective fashion from a laboratory's records; thus, information pertaining to many variables was not available for the HC. Hence, more studies are warranted to validate the results of comparisons between pwMS and the HC.

## Conclusion

The present study provides a descriptive overview of the gynecological health and related factors in a group of DMT-exposed wwMS living in a low-income setting. Results could aid future policy making and research. Furthermore, although Pap test results revealed a two-fold-increased risk of cervical cytological changes of malignant potential in DMT-exposed wwMS compared to HC, this difference did not reach statistical significance. Additionally, compared to matched HC, a significantly higher proportion of wwMS had evidence of infection in their Pap tests, while a significantly lower proportion of them showed benign inflammatory/reactive cellular changes. These findings are in support of a DMT-mediated immune dysfunction in the cervices of wwMS, the mechanistic background of which remains to be investigated.

## Acknowledgments

This work is based on a thesis for partial fulfilment of the MD degree.

## Author contributions

**Conceptualization:** Nahad Sedaghat.

**Data curation:** Shima Shoeib, Nahad Sedaghat.

**Formal analysis:** Nahad Sedaghat.

**Funding acquisition:** Masoud Etemadifar.

**Investigation:** Shima Shoeib.

**Methodology:** Nahad Sedaghat.

**Project administration:** Masoud Etemadifar.

**Resources:** Masoud Etemadifar, Mehri Salari, Mohammadreza Etemadifar.

**Software:** Nahad Sedaghat.

**Supervision:** Masoud Etemadifar.

**Validation:** Shima Shoeib, Nahad Sedaghat.

**Writing – original draft:** Nahad Sedaghat.

**Writing – review & editing:** Shima Shoeib, Mehri Salari, Mohammadreza Etemadifar, Nahad Sedaghat.

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
