## [Decision Letter · Decision Letter 0]

22 Nov 2024

PONE-D-24-46307Gynecologic health of women with multiple sclerosis: An overview on the current status and findings of Pap tests in a low-income settingPLOS ONE

Dear Dr. Sedaghat,

Thank you for submitting your manuscript to PLOS ONE. After careful consideration, we feel that it has merit but does not fully meet PLOS ONE’s publication criteria as it currently stands. Therefore, we invite you to submit a revised version of the manuscript that addresses the points raised during the review process.

We look forward to receiving your revised manuscript.

Kind regards,

Mohammad Reza Fattahi, M.D., M.P.H.

Academic Editor

PLOS ONE

**Journal Requirements:**

3. We note that your Data Availability Statement is currently as follows: All relevant data are within the manuscript and its Supporting Information files in aggregate form. 

**Additional Editor Comments:**

Thank you for submitting your manuscript entitled "Gynecologic health of women with multiple sclerosis: An overview on the current status and findings of Pap tests in a low-income setting." You have selected a highly relevant and underexplored topic that holds significant importance in the literature.

However, after a thorough review, I have identified several essential aspects that need to be addressed to enhance the clarity, depth, and impact of your manuscript. Please refer to the detailed comments and suggestions provided to guide your revisions. Addressing these points will significantly strengthen the manuscript's contribution to the field.

Reviewers' comments:

Reviewer's Responses to Questions

**Comments to the Author**

1. Is the manuscript technically sound, and do the data support the conclusions?

Reviewer #1: Yes

Reviewer #2: Yes

2. Has the statistical analysis been performed appropriately and rigorously? 

Reviewer #1: Yes

Reviewer #2: Yes

3. Have the authors made all data underlying the findings in their manuscript fully available?

Reviewer #1: Yes

Reviewer #2: No

4. Is the manuscript presented in an intelligible fashion and written in standard English?

Reviewer #1: Yes

Reviewer #2: Yes

5. Review Comments to the Author

**Reviewer #1: **This manuscript titled; Gynecologic health of women with multiple sclerosis: An overview on the current status and findings of Pap tests in a low-income setting

Thank you for your valuable contribution to the field. Early diagnosis of gynecological issues is critical for patients with MS, and the authors have provided an analysis to highlight this in their tables.

To further enhance the value of this manuscript, I would like to suggest additional points to help strengthen it.

Below are my comments for improvement:

Methods:

The authors haven't reported the total number of pregnancies in patients with MS, which may have a significant impact on the course of MS, particularly concerning the EDSS and disease progression. However, the manuscript does not specify the MS subtypes in the patients, such as relapsing-remitting (RR), primary progressive (PP), or secondary progressive (SP) MS. This lack of stratification is important, as the type of MS could influence both disease progression and potentially the results of Pap smears. It would be valuable for the authors to include an analysis of the MS subtype with Pap smear results, as this could reveal important patterns or associations. I would also suggest including a detailed analysis of the relationship between MS subtype and the Pap smear results, as this would offer more insight into the possible interactions between disease type and cervical health.

Additionally, in the exclusion criteria, patients with MS who have had intercourse within 48 hours of the Pap smear should be excluded from the study. This recommendation is based on established guidelines in gynecological research, as recent intercourse may interfere with test accuracy by causing irritation or altering the cells on the cervix. The authors should address this consideration more explicitly in the study design.

Tables:

Each table should have a clear and concise title placed above the table. Additionally, all abbreviations used within the tables should be listed below the table for clarity and ease of reference.

The percentages in each column should add up to exactly 100%, with no exceptions, to ensure accuracy in data presentation.

Furthermore, demographic data for the HC group are currently missing, I mean that all of the demographic data for MS patients in the first table must be provided for the HC group, too. This information is essential for making meaningful comparisons between the MS patients and the HC group. Including demographic data for both groups in the tables and within the body of the manuscript would enhance the comprehensibility and rigor of the study's findings.

**Reviewer #2: **1. Some claims made in the discussion and introduction sections are unsupported by references. Please provide adequate citations to strengthen the credibility of these statements. For instance:

Introduction:

*For instance, HPV vaccination could be offered to the wwMS if they are shown to bear an additional risk for HPV infection. Other instances could be offering behavioral consultation, sanitary products, as well as routine screening for, and treating of infections in both wwMS and their partners, if they are shown to bear an additional risk for such diseases.

Discussion:

*Interestingly, a recent Mendelian randomization study on the UK biobank cohort

found an association between MS and related factors (e.g., use of DMT) with cervical cancer but none of the other 14 investigated cancers.

*These results may be explained by an impairment in the immune response to cervical pathogens, possibly mediated by DMT.

*Furthermore, higher-grade inflammatory changes were more frequent in wwMS treated with aCD20 – a DMT class acting mainly on B cells.

*Another explanation could be the impairment of afferent pathways in wwMS. Particularly, impairment of afferent pathways in the wwMS with infection could render them asymptomatic.

2.About income:

*What is the methodology or scale for classifying income levels into mild, moderate, or severe categories?

*One critical aspect that required evaluation was the correlation between a patient's income level and their genital health. Unfortunately, this analysis is absent from the results and discussion sections. The study's methodology primarily just focused on matching MS patients with a control group for comparative analysis. → (Pap test results of wwMS were compared with a 1:2 age- and socioeconomic status (SES)-matched group of healthy controls (HC) who underwent Pap tests for cervical cancer screening in the same laboratory and in the same period)

3. Please specify in the methods section who provided the Pap smear samples: a gynecologist or another healthcare professional. This information is crucial for ensuring the adequacy and quality of the samples.

4. In certain instances please provide both the numerical count and the corresponding percentage together.

5. To enhance relevance, consider relocating this sentence ((Further of note, among all, only one of the wwMS with evidence of yeast infection showed severe inflammatory changes, whereas all HC with evidence of infection showed severe inflammatory changes)) and adding to the subsequent sentence ((As seen, the observed reactive/inflammatory changes differed between the participants with MS and matched HC; such changes were less frequently observed in the wwMS (OR: 0.3; 95% CI: 0.2, 0.4; p < 0.001)).

6. When analyzing (Pap test results), it would be insightful to rank DMTs based on their associated odds ratios (ORs), from highest to lowest. This ranking would facilitate the identification of DMTs potentially linked to increased risk of infection or inflammatory. changes in MS patients. Also for better understanding you can mention as sentences which of DMT can cause more infection or inflammatory reaction?

7. In paragraph 3 of the discussion, please add the word (neural) before (afferent pathway) for better understanding.

8. please add (Benign) word before all phrases (inflammatory/reactive changes) for more comprehension.

9. As comorbidities such as diabetes can significantly impact susceptibility to infection in MS patients, it is crucial to explore the relationship between specific comorbidities and the risk of infection in wwMS. please explain the relationship or impact of comorbidities on infection in wwMS.

10. please mention the mean (SD) or median of duration of DMT.

11. Please provide more detailed information on the prevalence of Human Papillomavirus (HPV) infection in wwMS. If the prevalence of HPV is elevated compared to the general population, please elucidate the potential factors contributing to this increased risk. Additionally, please discuss the cervical malignancy presentation associated with HPV infection in MS patients.

6. PLOS authors have the option to publish the peer review history of their article (what does this mean?). If published, this will include your full peer review and any attached files.

Reviewer #1: No

Reviewer #2: No

---

## [Author Response · Author response to Decision Letter 1]

4 Jan 2025

Journal Requirements

Response: We ensured the revised files meet the style requirements as described in the mentioned templates.

Response: The contact information for the respective ethics committee was added in the revised manuscript. The Data Availability statemen was updated in the manuscript and submission form accordingly.

3. We note that your Data Availability Statement is currently as follows: All relevant data are within the manuscript and its Supporting Information files in aggregate form.

Response: As mentioned, the manuscript contains all relevant “aggregate” data, and does not include “raw” data. The raw data contain potentially identifying information pertaining to the participants of the study. Thus, the contact information for the respective ethics committee was added in the revision. The Data Availability statemen was updated in the submission form accordingly. 

Additional Editor Comments:

“Thank you for submitting your manuscript entitled "Gynecologic health of women with multiple sclerosis: An overview on the current status and findings of Pap tests in a low-income setting." You have selected a highly relevant and underexplored topic that holds significant importance in the literature.

However, after a thorough review, I have identified several essential aspects that need to be addressed to enhance the clarity, depth, and impact of your manuscript. Please refer to the detailed comments and suggestions provided to guide your revisions. Addressing these points will significantly strengthen the manuscript's contribution to the field.”

Response: Thank you for dedicating your time to our manuscript. We are confident that the revision has added to the publication merit of our manuscript.  

Reviewer's Responses to Questions

1. Is the manuscript technically sound, and do the data support the conclusions?

Reviewer #1: Yes

Reviewer #2: Yes

Response: We would like to thank both of the reviewers for acknowledging the technical soundness of our study and the appropriateness of the conclusions.

2. Has the statistical analysis been performed appropriately and rigorously?

Reviewer #1: Yes

Reviewer #2: Yes

Response: We would like to thank both of the reviewers for acknowledging the appropriateness and rigor of the statistical methods.

3. Have the authors made all data underlying the findings in their manuscript fully available?

Reviewer #1: Yes

Reviewer #2: No

Response: The data produced during the presented study is presented in aggregate form. Raw data is not included in the submission. The raw data contains potentially identifying information pertaining to the participants of the study, thus, its public, unrestricted sharing may raise ethical issues. We added the contact details of the research ethics committee to the revised manuscript.

4. Is the manuscript presented in an intelligible fashion and written in standard English?

Reviewer #1: Yes

Reviewer #2: Yes

Response: We would like to thank both of the reviewers for acknowledging the clarity of the manuscript’s language.  

Review Comments to the Author

Reviewer #1

“This manuscript titled; Gynecologic health of women with multiple sclerosis: An overview on the current status and findings of Pap tests in a low-income setting

Thank you for your valuable contribution to the field. Early diagnosis of gynecological issues is critical for patients with MS, and the authors have provided an analysis to highlight this in their tables.

To further enhance the value of this manuscript, I would like to suggest additional points to help strengthen it.”

Response: Thank you for dedicating your time to our manuscript. We endeavoured to address your comments in a clear and comprehensive fashion, and believe doing so has improved the quality of our manuscript significantly.

“Below are my comments for improvement:

Methods:

The authors haven't reported the total number of pregnancies in patients with MS, which may have a significant impact on the course of MS, particularly concerning the EDSS and disease progression.”

Response: Thank you for this comment. Indeed, each participant’s pregnancies could have affected MS outcomes. As our outcome was not disability worsening, we regretfully did not collect information regarding the number of prior pregnancies of participants. We explicitly acknowledged this point as a limitation so that future studies could bear this particular point in mind (limitations paragraph in discussion section).

“However, the manuscript does not specify the MS subtypes in the patients, such as relapsing-remitting (RR), primary progressive (PP), or secondary progressive (SP) MS. This lack of stratification is important, as the type of MS could influence both disease progression and potentially the results of Pap smears.”

Response: Thank you for your comment. The third line of the first paragraph of the “results: characteristics of participants” section reads:

“At the time of the study, 18 (9.1%) had secondary progressive, while the others had relapsing-remitting MS.”

As interpreted, no person with primary progressive MS participated in the study. This was mentioned as a limitation in the revised manuscript (limitations paragraph in discussion section).

“It would be valuable for the authors to include an analysis of the MS subtype with Pap smear results, as this could reveal important patterns or associations. I would also suggest including a detailed analysis of the relationship between MS subtype and the Pap smear results, as this would offer more insight into the possible interactions between disease type and cervical health.”

Response: Thank you for allowing us to clarify. The last paragraph of the “3.3. Pap test results” subsection of the “Results” section reads:

“Furthermore, using ordinal and binary logistic regression, the association of the pap test results (i.e., the outcome variables mentioned above) was investigated with the covariates age, MS duration and subtype, total number of MS relapses, total number of times undergoing corticosteroid pulse therapy, EDSS score, historical and current DMT and duration thereof. A positive association was found between current aCD20 therapy and the degree of benign inflammatory/reactive changes (unadjusted OR vs. interferons: 2.7; 95% CI: 1.1, 6.8; p = 0.03). This association was seen multivariable analysis as well (adjusted OR vs. interferons: 3.5; 95% CI: 1.3, 9.1; p = 0.01). Other results returned unremarkable.”

 As mentioned, MS subtype was already included as a covariate in the regression analyses, returning unremarkable results. This could be due to the fact that the study lacked people with primary progressive MS and had a low count of participants with secondary progressive MS. We acknowledged this in the revised discussion section (limitations paragraph in discussion section).

“Additionally, in the exclusion criteria, patients with MS who have had intercourse within 48 hours of the Pap smear should be excluded from the study. This recommendation is based on established guidelines in gynecological research, as recent intercourse may interfere with test accuracy by causing irritation or altering the cells on the cervix. The authors should address this consideration more explicitly in the study design.”

Response: Thank you for mentioning. The sixth inclusion criterion of the study reads:

“absence of any medical, cultural, and/or ethical contraindication for Pap testing.”

We acknowledge the ambiguity of this sentence; thus, the revised sentence now features examples of considered “contraindications” to enable most accurate comprehension by the readers.

“Tables:

Each table should have a clear and concise title placed above the table. Additionally, all abbreviations used within the tables should be listed below the table for clarity and ease of reference.”

Response: Thank you for your suggestion. The revised tables now feature a clear and concise title above, and a list of all abbreviations (in order of being mentioned in the table) below.

“The percentages in each column should add up to exactly 100%, with no exceptions, to ensure accuracy in data presentation.”

Response: Thank you for mentioning. We noticed that due to rounding of the percentages in some instances, the percentages in some columns adds up to 100% +/- 0.2% instead of exactly 100%. We revised the rounding so that all percentages add up to 100% in all tables. Additionally, the following revisions were made in Table 1: (i) the count and percentage for those not reporting any method for contraception was added, and erroneous percentages in this section were corrected; (ii) the count and percentage of ones reporting no comorbidity was added. Please note, some persons reported having multiple comorbidities, so the percentages in the comorbidities section correctly exceeds 100%. (iii) erroneous count and percentages in the “usage of vaginal cleansing products” section was corrected. In Table 2: (i) erroneous count of healthy controls without evidence of infection was corrected.

“Furthermore, demographic data for the HC group are currently missing, I mean that all of the demographic data for MS patients in the first table must be provided for the HC group, too. This information is essential for making meaningful comparisons between the MS patients and the HC group. Including demographic data for both groups in the tables and within the body of the manuscript would enhance the comprehensibility and rigor of the study's findings.”

Response: Thank you for mentioning your concern. As mentioned in the manuscript, the healthy control group consisted of people who referred to the same laboratory in the same period; the data of whom was collected retrospectively from the laboratory records. The laboratory records featured basic data to confirm eligibility of the healthy controls (e.g., age, socioeconomic status, symptoms, reason for referral, etc.), yet, at the time of the study, we could not gain access to their contact information. Thus, the healthy controls were not surveyed, and the survey data (i.e., information presented in Table 1) is not available for the healthy controls. This was explicitly stated as a limitation (limitations paragraph in discussion section); we are hoping future investigators account for this point through reading our manuscript.  

Reviewer #2

 “1. Some claims made in the discussion and introduction sections are unsupported by references. Please provide adequate citations to strengthen the credibility of these statements.”

Response: Thank you for noticing, and apologies for overlooking such errors. We made sure all the references are added to the manuscript.

“For instance:

Introduction:

*For instance, HPV vaccination could be offered to the wwMS if they are shown to bear an additional risk for HPV infection. Other instances could be offering behavioral consultation, sanitary products, as well as routine screening for, and treating of infections in both wwMS and their partners, if they are shown to bear an additional risk for such diseases.”

Response: References were added.

“Discussion:

*Interestingly, a recent Mendelian randomization study on the UK biobank cohort found an association betwe

---

## [Decision Letter · Decision Letter 1]

13 Feb 2025

Gynecologic health of women with multiple sclerosis: An overview on the current status and findings of Pap tests in a low-income setting

PONE-D-24-46307R1

Dear Dr. Sedaghat

We’re pleased to inform you that your manuscript has been judged scientifically suitable for publication and will be formally accepted for publication once it meets all outstanding technical requirements.

Kind regards,

Mohammad Reza Fattahi, M.D., M.P.H.

Academic Editor

PLOS ONE

Additional Editor Comments (optional):

Reviewers' comments:

Reviewer's Responses to Questions

**Comments to the Author**

1. If the authors have adequately addressed your comments raised in a previous round of review and you feel that this manuscript is now acceptable for publication, you may indicate that here to bypass the “Comments to the Author” section, enter your conflict of interest statement in the “Confidential to Editor” section, and submit your "Accept" recommendation.

Reviewer #2: All comments have been addressed

2. Is the manuscript technically sound, and do the data support the conclusions?

Reviewer #2: Yes

3. Has the statistical analysis been performed appropriately and rigorously? 

Reviewer #2: Yes

4. Have the authors made all data underlying the findings in their manuscript fully available?

Reviewer #2: Yes

5. Is the manuscript presented in an intelligible fashion and written in standard English?

Reviewer #2: Yes

6. Review Comments to the Author

Reviewer #2: The manuscript was excellently revised. Only two minor points are considered:

1. In the abstract: Please expand (women with MS (wwMS) to (women with multiple sclerosis)

2. Please put the point after the reference number at the end of sentences in your manuscript : .[] to [].

7. PLOS authors have the option to publish the peer review history of their article (what does this mean?). If published, this will include your full peer review and any attached files.

Reviewer #2: No

---

## [Editor Report · Acceptance letter]

PONE-D-24-46307R1

PLOS ONE

Dear Dr. Sedaghat,

I'm pleased to inform you that your manuscript has been deemed suitable for publication in PLOS ONE. Congratulations! Your manuscript is now being handed over to our production team.

Kind regards,

on behalf of

Dr. Mohammad Reza Fattahi

Academic Editor

PLOS ONE